# Hive Orientation and Colony Strength Affect Honey Bee Colony Activity during Almond Pollination

**DOI:** 10.3390/insects15020112

**Published:** 2024-02-05

**Authors:** Sandra Kordić Evans, Huw Evans, William G. Meikle, George Clouston

**Affiliations:** 1Canetis SRL., 55022 Bagni di Lucca, Italy; 2Beehero Inc., Tel Aviv 6721117, Israel; huw@beehero.io (H.E.); george@beehero.io (G.C.); 3Carl Hayden Bee Research Center, United States Department of Agriculture-Agricultural Research Service (USDA-ARS), Tucson, AZ 85719, USA; william.meikle@usda.gov

**Keywords:** almond pollination, hive orientation, hive strength, foraging activity, bee counter, hive weight

## Abstract

**Simple Summary:**

Managed honey bees are virtually exclusive pollinators of almonds in commercial agricultural contexts such as that of California’s Central Valley where 80% of the world’s almonds are produced. Streamlining almond pollination by utilizing honey bee colonies in the most efficient way aims to support both the growers and beekeepers as demand for pollination services and per unit costs are increasing. This study examines whether hive entrance orientation and hive strength influence the pollinating activity of honey bees in almond orchards. We have found that hives facing east have a marked advantage in starting their activity in the morning compared to hives facing west. Similarly, strong colonies show a significantly earlier start of activity than weak colonies. The practical implication of these findings lies in configuring the placement of colonies of varying strengths in a way that favours their foraging activity.

**Abstract:**

The foraging activity of honey bees used to pollinate almonds was examined in relation to their hive entrance orientation and colony strength. Twenty-four colonies of honey bees, twelve in each group, were situated with their entrances facing east and west cardinal points. Bee out counts were recorded continuously and hive weight data at ∼10 min intervals from 17 February to 15 March 2023. Colony strength was assessed using the frames of adult bees (FOB) metric. East-facing hives started flight 44.2 min earlier than west-facing hives. The hive direction did not affect the timing of the cessation of foraging activity. The hive strength played a significant role: hives assessed as weak (≤3.0 FOB) commenced foraging activity 45 min later than strong hives (>3.0 FOB) and ceased foraging activity 38.3 min earlier. Hive weight data did not detect effects of either the hive direction or colony strength on the commencement and cessation of foraging activity, as determined using piecewise regression on 24 h datasets. However, the hive weight loss due to foraging activity at the start of foraging activity was significantly affected by both direction (East > West) and colony strength (Strong > Weak). Our study showed that, during almond pollination, both hive entrance exposure and hive strength have quantifiable effects on colony foraging behaviour and that these effects combine to regulate the overall foraging activity of the pollinating colonies.

## 1. Introduction

Almonds grown in the Central Valley of California account for 80% of the world’s production [1] and their global annual value is estimated at several billion dollars [2]. Most varieties of almonds grown in California are self-incompatible and require the entomophilous transfer of pollen between compatible cultivars [3,4,5]. Managed honey bee colonies are by and large the exclusive pollinators of almonds in the Central Valley of California due to a lack of semi-natural habitats that could support non-*Apis* pollinators [6]. It is estimated that pollination services now account for 20% of the total annual operating costs for almond growers [7,8,9]. The value of the pollination services provided is thus one of the major concerns for stakeholders in the industry, growers and beekeepers alike. Streamlining the effectiveness of pollination services offers the potential to impart benefits to all involved as well as providing further understanding of honey bee behaviour in this ever-increasing agricultural input.

Honey bee foraging behaviour in almonds and how it influences pollination success has been the subject of numerous studies, which have shown to what extent endogenous and exogenous factors modulate the activity of pollinating units [10,11,12]. It is generally considered that temperatures below 12 °C, winds stronger than 24 km/h and rainfall prevent bee foraging activity. Decreased solar radiation due to cloud cover also reduces bee activity. Furthermore, strong colonies are thought to be more active than weak colonies, and this is more prominent at the aforementioned threshold environmental values [4,10]. For this reason, colony strength is an important part of almond pollination contracts: rented colonies in general must meet some minimum size of the adult bee population, and a bonus is provided for strong colonies well in excess of that minimum size. While these parameters form a solid predictive base for pollination efficiency, there are a number of other factors that could modulate honey bee activity during almond pollination. Due to the fact that environmental conditions during the almond bloom in California are unpredictable and can pivot around threshold climatic values, we investigated to what extent the orientation of the hive entrance with respect to sunrise and sunset affects the honey bee foraging activity and potential pollination success.

General beekeeping advice on the placement of honey bee colonies in the Northern hemisphere suggests that hive entrances should face south-east [13] or have a southern or eastern exposure [14]. This orientation favours better productivity and survival, as it maximises the exposure to solar radiation, thus stimulating flight earlier in the day, during the season and, during the colder months, it may make a difference as to whether the bees leave the hive for cleansing flights. This is also supported by the work on bait hives [15], which demonstrated swarms’ preference for southerly exposure, although another study on wild nests showed a random distribution of hive nest entrance directions [16]. In contrast, a study of 108 feral honey bee nests found the prevalence of SW-facing entrances compared to any other direction [17]. It is worth mentioning that a number of other beekeeping guides advise arranging the colonies in an apiary with entrances pointing in diverse directions so as to avoid drifting [18,19]. This consideration places the emphasis on mitigating the negative effects of drifting, namely, the unbalanced loss or gain of workers, as well as favouring the spread of disease. In pollination scenarios, it is recommended that hive entrances be exposed to early morning sun, hence east or southeast, to stimulate early foraging, which is of particular importance in certain crop cultures [20].

Successful almond pollination, which occurs at the end of the winter and entails a sufficient transfer of pollen between compatible cultivar trees, is presented with a number of challenges; climatic conditions, which are often limiting for honey bee flight and honey bee colonies physiologically at their weakest in terms of population, are the two major factors [3]. Unlike the climate, which cannot be controlled, honey bee pollination activity can be influenced to a degree by beekeeper’s management practices that ensure strong colonies. This study was designed to assess the effects that hive orientation and hive strength have on the pollination activity of the colonies once in the orchard. There is comparatively little published data on the effects of hive orientation on honey bee activity; Meikle et al. [21] used hive weight data to assess the foraging activities of colonies facing four cardinal points in Arizona in 2019 and 2020. That work showed that colonies facing east consistently started flying earlier in the months of January to April and that this difference disappeared as the season progressed and limitations to flight by climatic conditions dissipated.

Electronic equipment to monitor honey bee colonies and their activity that has been deployed in recent years has provided a higher resolution insight into colonies’ behaviour and offers the potential to make management decisions for more efficient pollination. In this study, data from entrance activity sensors (bee counters) and hive scales were used to evaluate the honey bee foraging behaviour in two groups of colonies whose entrances faced opposing cardinal points, east and west. Previous work on bee counters has demonstrated their utility [22,23]. Here we present the results collected during the 2023 almond bloom in the Central Valley and discuss the possible implications of the hive position on the timing of foraging activity, total flying time and effects of the colony size on the time of leaving and the total amount of activity on the foraging behaviour patterns.

## 2. Materials and Methods

### 2.1. Study Site and Honey Bee Stock

The field study was conducted from 17 February to 15 March 2023, in a commercial almond orchard in Tranquility (Fresno County, CA, USA, 36.65029° N, 120.27688° W). The plot of 66 acres is surrounded predominantly by other almond orchards, a small section of field crops along the northern edge and idle land along the southern edge of the plot. The almond cultivars, spread over the entire orchard, were 50% Nonpareil, 25% Aldrich and 25% Monterey. The orchard was supplied with 132 colonies derived from Russian honey bees at a recommended stocking rate of 2 hives/acre. Eight of these Russian colonies were selected for this study. To obtain a degree of genetic diversity in the study, eight Pol-line colonies (obtained from USDA-ARS, Baton Rouge, LA, USA) and eight unselected Stock colonies (supplied by a beekeeper) were also installed at the site. The genetic backgrounds of the hives were not considered further here. The resulting 24 study colonies were positioned away from the rest of the pollinating units in a line along the access road at the south edge of the orchard and were placed on six pallets, each supporting four hives. Each of the 24 colonies was housed in two deep 10-frame Langstroth boxes. Half the hives had their entrances east-facing and the other four had the entrances west-facing, evenly distributed among queen lines.

### 2.2. Electronic Equipment

All 24 hives were fitted with entrance activity counters (bee counter, Figure 1), a custom-built optical sensor that continuously collects the data on honey bees exiting the hive. Bees leaving the hive pass through one of the bee counter’s 26 tunnels. Each tunnel has two infrared light beams; as a bee moves through the tunnel, these light beams are broken, incrementing the count of bees exiting the hive. This measure is used to assess the foraging activity of the colony. Although not all bees exiting the hive are foragers, for example, those on cleansing and orientation flights, this overestimation applies to all colonies and is unlikely to affect the data in a significant way. The sensor accuracy was established to be 97% (unpublished data) in previous tests performed using the “Robbers test” methodology described by Struye [24].

Weight data for each hive were collected using the custom-built electronic hive scales, (stainless steel doughnut design comprising four load cells, one in each corner of the scale; max load capacity: 160 kg, precision: ±100 g). The meteorological conditions were continuously monitored throughout the duration of the experiment using a purpose-built weather station, composed of a rain gauge, i.e., a tipping bucket sensor that measures rainfall, a spinning cups sensor that measures the wind speed and a wind vane sensor that determines the wind direction. A thermometer that measures ambient temperature was housed within the rain gauge enclosure. All the weather sensors were mounted to a stand, which was placed at one of the central pallets of hives as seen in Figure 2. Power was supplied to the equipment fitted to each hive using a 5 W solar panel and a lithium-ion power bank, housed in a waterproof enclosure, as seen on top of the hives (circled blue in Figure 2). All data measurements from the weather station, the weight registered by the hive scales and the cumulative bee activity measured by the bee counter were recorded synchronously at 10 min intervals and uploaded to the central server every 30 min, using a communication gateway connecting to the cellular network (circled red in Figure 2).

### 2.3. Hive Strength and Activity

All hives were subject to a colony strength assessment on three occasions during the experiment; (17 February 2023, 22 February and 17 March 2023). The first two inspections were completed using the Liebefeld method [25] to assess the frames occupied by the adult bees, eggs, open and closed brood, nectar and pollen stores, whereas the third was a generic beekeeper estimation of the frames occupied by the adult bees and nectar stores. Data on colony strength, expressed as frames of adult bees (FOB), were analysed with respect to the treatment group (east- and west-facing entrances). The daily flight commencement and cessation times were determined as a timestamp at which the number of bees exiting the hive was ≥10 bees and <10 bees per minute (BPM), respectively. Subsequently, the bee flight hours were calculated as a sum of the times within range of the aforementioned BPM threshold parameters. Within each treatment group, the colonies were divided into two subgroups, weak and strong, based on the FOB parameter. The weak colony group consisted of six colonies with strengths of ≤3.25 FOB, whereas the strong colony group comprised six colonies of >3.25 FOB.

### 2.4. Statistical Analysis

Adult bee population estimates were subjected to a repeated measures MANOVA (Proc Glimmix, SAS Inc., Cary, NC, USA, 2002) to evaluate the fixed effects of the direction (East vs. West) and days after the start of the experiment (5 and 23), with the hive number as a random effect and with the estimate on the first day as a covariate to the control for pre-existing differences.

Bee count data were subjected to a repeated measures MANOVA to evaluate the fixed effects of direction (East vs. West), colony strength (Strong, or FOB > 3.0 vs. Weak, or FOB ≤ 3.0), day, and all two-way interactions, on the daily estimates of colony start and stop times (expressed as proportion of day). The hive number was used as a random effect, an ar(1) autoregressive covariance structure was used, and the degrees of freedom were calculated using the Kenward-Roger method.

Continuous hive weight data per day (from midnight to midnight) were fitted with a piecewise regression using R Studio 2022.07.2 (RStudio, PBC, Boston, MA, USA, 2022), with 10 iterations per day sample, yielding estimates for 4 break points, 5 slope values and the adjusted r^2^ [26]. Three parameters were used in the statistical analyses: (1) the dawn break point t_D_ (start of daily foraging activity); (2) the dusk break point t_D_ (start of daily foraging activity); and (3) the slope of the 1st segment after dawn S_M_ (the rate of morning hive weight change). S_M_ was attributed to both forager departure and moisture loss (i.e., nectar drying and respiration). To better estimate weight change due to forager departure, the effect of moisture weight change was removed as follows:ΔF = S_M_(t_D+1_ − t_D_) − S_N_(t_D+1_ − t_D_)(1)
with ΔF = the hive weight change due to the forager departure; S_N_ = the night slope (the rate of hive weight change due to moisture loss from midnight until dawn)_;_ and t_D+1_ = the break point following the dawn break point [27]. ΔF, t_D_ and the dusk break point were used as response variables in the MANOVA analyses (see above). Finally, the hive weight change (g) every 24 h from midnight to midnight, divided by the colony FOB (to control for colony strength) was used in another MANOVA analysis with the same fixed effects as above.

## 3. Results

### 3.1. Colony Strengths

While there was considerable variation in strength within each group (Figure 3), the average strength of each group was almost equal and, furthermore, remained comparable throughout the course of the study. This finding allowed for an unbiased comparison of the foraging activities of the two groups as the strength of the colony is expected to influence the foraging activity profile and intensity. No effect of orientation was observed on the estimated adult bee populations during the experiment (*p* = 0.79).

### 3.2. Flight Start and End Times with Respect to Orientation and Hive Strength

The MANOVA results for the start and stop daily response variables are shown (Table 1). For the start time, all main effects and interactions are significant. Using the least-squares mean differences from the statistical output, hives facing west started foraging on average 0.0307 days, or 44.2 min, later than hives facing east. Likewise, the weak hives started foraging on average 0.0313 days, or 45.0 min, later than the strong hives; the daily profiles are depicted in Figure 4. The stop time was not significantly affected by hive direction, but the effect of the hive strength was significant: strong hives ceased foraging activity on average 0.0266 days, or 38.3 min, later than weak hives (Figure 5). The weather (Figure 6) apparently played a role in some data—days with rainfall had distinctly later start times and earlier end times than other days. On the 24th and 25th of February, low ambient temperatures coupled with rainfall accounted for no flight activity in the west-facing weak colonies and greatly reduced activity in all other groups.

### 3.3. Total Foraging Trips and Flight Hours (Density of Flights)

Daily bee flights were subdivided into one-hour intervals between 07:00 and 19:00, and the total number of trips in each time slot was analysed (Figure 7). The total number of trips for east-facing colonies was significantly greater than for west-facing colonies between 08:00 and 10:00. Subsequently, the difference in flights between groups diminishes progressively throughout the central hours of the day, and finally, in the late afternoon, the west-facing colonies show higher flight activity than the east-facing colonies, although this difference is not statistically significant.

In order to analyse whether this starting advantage in the east-facing group had an effect on the overall foraging activity of the two groups, the cumulative daily foraging trips for both groups were compared and statistically evaluated. Data on the first and last day of the study as well as two days of rain were excluded from the calculations, as they were incomplete. Twelve colonies in the east-facing group made 5,307,899 foraging trips during the 24-day period, whereas the twelve colonies in the west-facing group made 4,563,880 foraging trips. Although the total number of trips is marginally greater in the east-facing group, this difference shows no statistical significance (*p* = 0.29).

Dividing the cumulative daily trip figure of each group by the number of hours during which the bees egressed from the colonies yielded a measure of the flight density. The number of foraging trips per hour for the east-facing group is 30,859/h and 34,587/h for the west-facing group. Whereas flight density is marginally higher in the west-facing group, the results were not statistically significant (*p* = 0.31).

### 3.4. Weight Data

Piecewise regressions of the hive weight data were used to determine three parameters: (1) the start of foraging activity in the morning (“dawn break point”); (2) the end of foraging activity in the evening (“dusk break point”); and (3) the size of the foraging population, based on the hive weight loss just after the start of the foraging activity. Only the foraging activity was found to be affected by either the hive direction or colony strength (Table 2). The dawn and dusk break points clearly changed over time, but either the scales were not sensitive enough to reliably detect those points and/or the piecewise regression method used here, applying a four-break point line over the entire 24 h dataset, was not sensitive enough to accurately detect the desired parameters (see Figure 8). The “forager population” estimate, based on the slope of the segment after the dawn break point, was significantly affected by both the hive direction and colony strength. The east-facing hives showed an initial foraging weight loss of 50 g more on average than west-facing hives, based on the least-squares mean differences, and the strong colonies showed an initial foraging weight loss of 35 g more on average than the weak colonies. The hive weight change from midnight to midnight, divided by the FOB per colony, was not affected by either the hive direction (*p* = 0.51) or colony strength (*p* = 0.74).

## 4. Discussion

Our data, using exiting bee counts, showed that east-facing colonies start flying 44.2 min earlier than west-facing colonies. This is in close agreement with previous work [21] that found that east-facing colonies started daily flight activity 50 min earlier than west-facing colonies. The difference in activity thus presents a potential advantage to the pollination efficiency of east-oriented colonies compared to west-oriented colonies, specifically in crop situations where pollen availability may become limited due to diurnal patterns of production, as in almonds, where pollen is released diurnally once temperatures reach 13 °C and, under favourable foraging conditions, is removed by the bees by early afternoon [28,29]. On days when low temperatures and precipitation in the morning prevented honey bee foraging activity, there were no net differences in start times of the east- and west-facing groups. Hive orientation on those days did not appear to affect the pollination activity of the two groups, as the advantage of the early morning sunlight in east-oriented hives is absent.

We found no real difference between the east- and west-facing colonies ending their daily flight activity. Interestingly, both west- and south-facing colonies in the study during December–March 2019 ended their flight activity significantly later than east-facing colonies [21]. One possible explanation for the lack of differences in the activity of east- and west-facing hives in our study is that the ambient temperatures at the end of the day were above the threshold values on most days; hence, the temperature was not a limiting factor to flight. Moreover, the availability of a high-reward pollen resource was possibly the main driver behind the colonies’ activity. A study assessing the availability of pollen in trees would be required to evaluate this postulate, and it would be particularly interesting as it is thought that, when the ambient conditions are favourable and the pollination colonies strong, the pollen on almond flowers is stripped by the early afternoon.

Our data for the whole-day activity, averaged for each of the two groups, show that despite a significant difference in start times, the overall flight activity between the two groups is comparable, as the west-facing hives increase the density of flying in the later hours of the day, neutralising the apparent advantage of the east-facing colonies’ early start. However, the greater flight density of the west-facing colonies is to some extent a function of their shorter daily flight hours. It is worth noting that despite the lack of a significant difference in the flight activity of the two groups, both in terms of total flights and flight density, it is apparent that west-facing colonies do exhibit marginal yet consistently lower foraging activity. Whether this marginal difference in pollination activity conveys a tangible benefit to crop yield is to be further explored over multiple seasons and larger group data sets.

A number of studies have shown that stronger colonies make more flights and collect more pollen than weaker ones [30,31]. Over the decades, these studies coupled with anecdotal evidence have resulted in the almond industry advocating the use of strong colonies to ensure maximum pollination. By examining the bee count flight activity data not only with respect to orientation but also to colony strength, we found that weak colonies facing west are at a strong disadvantage in terms of early morning flight activity in contrast to comparable colonies facing east. From this information, it is evident that, at threshold ambient conditions, weaker colonies strongly benefit from being exposed to early solar radiation. While it may not be practical to configure the pollination units to all face a favourable east or possibly south direction, it may be worth considering, within the constraints of the colony placement, orienting weaker hives to receive the early morning sun. Furthermore, we have found that weak colonies, regardless of their orientation, started foraging 45 min later and ceased foraging 38.3 min earlier compared to strong colonies, adding to an average of over an hour less foraging time per day. These findings support the use of stronger colonies as not only more active pollinators but also as more resilient ones in marginal environmental conditions, such as the winter pollination of almonds. The simplest explanation for stronger colonies having an earlier start and later stop times could merely lie in greater numbers of bees exiting or entering the hive during optimal traffic periods. The criterion for the start and stop of the foraging activity, 10 bees per minute, is not a function of the colony size, so one would expect that the larger the colony, the earlier it would meet that criterion in the morning, and the later it would meet it in the afternoon, even if the mean departure time was the same as that of a smaller colony.

The dawn and dusk break points, derived from the hive weight data, which denote the commencement and cessation of bee activity, did not show the effect of the hive orientation and strength on flight activity that was clearly seen in the bee counter data. This is most likely explained by the fact that the sensitivity of the hive scales (±100 g, which equates to ∼1000 bees) is much lower than the precision of the out counts recorded by the bee counter. Another factor is the fitting of the piecewise regressions to 24 h datasets. The disadvantage of using the full 24 h is that weight changes earlier or later in the day, which had little or nothing to do with foraging activity, played a role in the curve fit and thus may have reduced the overall sensitivity of the weight analysis to the parameters of interest.

While the dawn and dusk break points were not affected by direction or colony strength, the forager population mass parameter, also derived from hive weight changes, was affected. This suggests that the hive orientation and hive strength significantly affect the number of foraging bees exiting the colonies and confirms the out-count data. Interestingly, daily hive weight changes, when divided by the FOB data to control for the colony size, were unaffected by direction or colony strength. Assuming that the data quality (the precision of weight and FOB measures) did not play a role, this may be because, given the abundance of the forage at that time and place, the timing of foraging is not critical for food acquisition. In addition, while bigger colonies have been shown to be quicker at locating and recruiting resources than small colonies, which could explain how they mobilise foragers earlier in the day, as we detected, they have not been shown to recruit proportionately (rather than numerically) more foragers to abundant forage sources [32], potentially explaining the lack of the effect of the colony strength on weight changes per FOB unit.

It is, therefore, worth noting that, for the data on honey bee foraging activity, electronic bee counters and hive scales both offer viable solutions. Bee counters provide the advantage of a direct measure of the activity, whereas an inference is required from the hive scale data. The weight data may be advantageous in studies looking at foraging activity with respect to nectar collection and depletion of stores. Where feasible, we would recommend the use of both sensors as the two streams of data offer a synergy in analysis. Also, we examined the foraging parameters chosen at least in part for their ease of computation and comparison across sensor types. Other parameters, such as the length and distribution of the total foraging effort, may also be of interest in future work.

Our results show how the flight activity of pollinating honey bees is affected by hive strength and hive entrance orientation, which in turn could steer recommendations to growers and beekeepers. Beyond deploying strong colonies, growers could gain a better pollination service by optimising the placement of weaker colonies, although the logistics required to achieve this would be problematic in practice. Beekeepers, for whom almond pollination provides an opportunity to build up the colonies for the forthcoming season, could extract a greater benefit if the colony activity is optimised by a simple configuration consideration. While this particular study remains to be replicated across field seasons and expanded to include multiple orientations (S, N, SE, SW, NE, NW), the approach is novel in that foraging data were collected and compared from two independent sensor types—bee counters and hive scales—which are rarely completed in such field studies, and because these results largely confirmed those from a previous study, so with respect to the east and west entrance orientations, this study represents a replication of those findings. Finally, it is important at this point to reflect on the relatively unique situation in almond pollination. Almond pollination and its effectiveness are considered a community undertaking encompassing surrounding orchards, regardless of ownership, within pollinating honey bees’ flight range and, as such, any optimisation has effects reaching beyond a single orchard.

## Figures and Tables

**Figure 1 insects-15-00112-f001:**
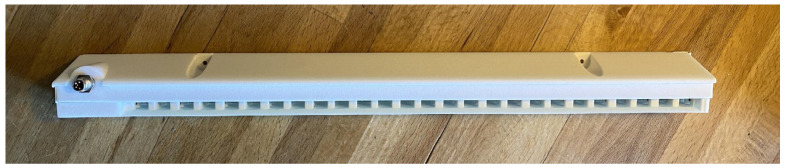
Optical entrance activity sensor (bee counter).

**Figure 2 insects-15-00112-f002:**
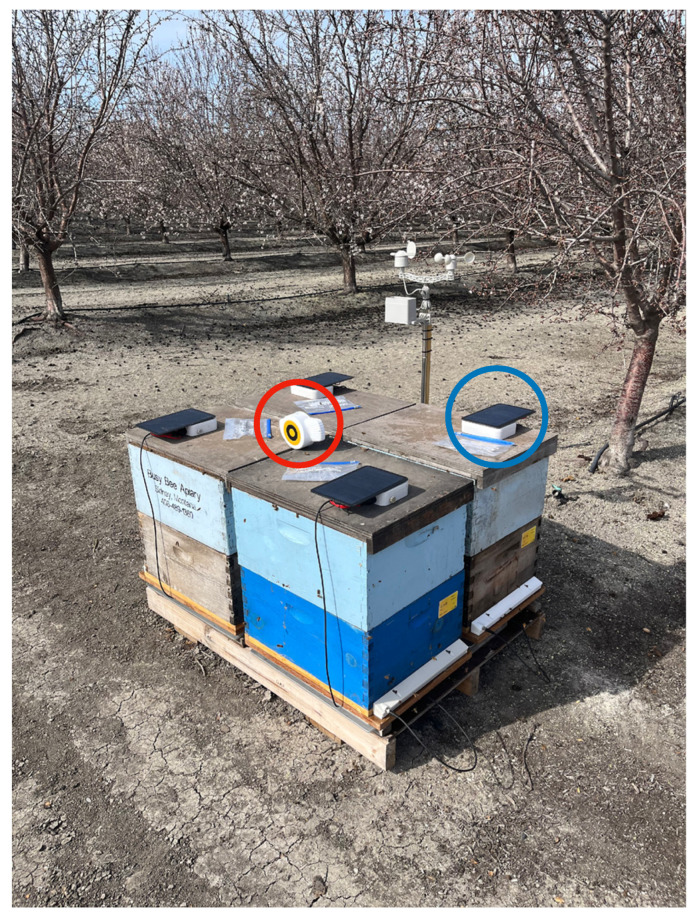
The bee counters, power supply, communication gateway and weather station mounted to a central pallet.

**Figure 3 insects-15-00112-f003:**
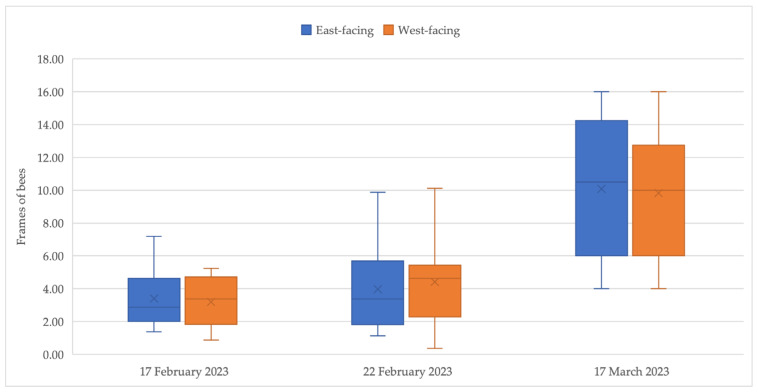
The colony strength (FOB) at three time points during almond pollination.

**Figure 4 insects-15-00112-f004:**
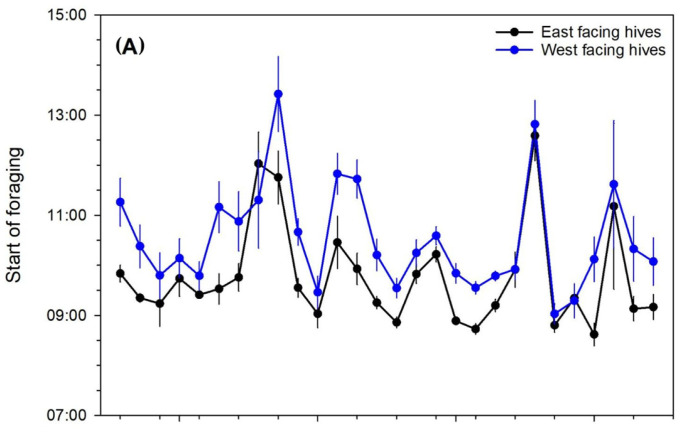
The average flight start time of the colonies during almond pollination for (**A**) east- vs. west-facing colonies and (**B**) strong vs. weak colonies. Error bars represent the standard error of the mean.

**Figure 5 insects-15-00112-f005:**
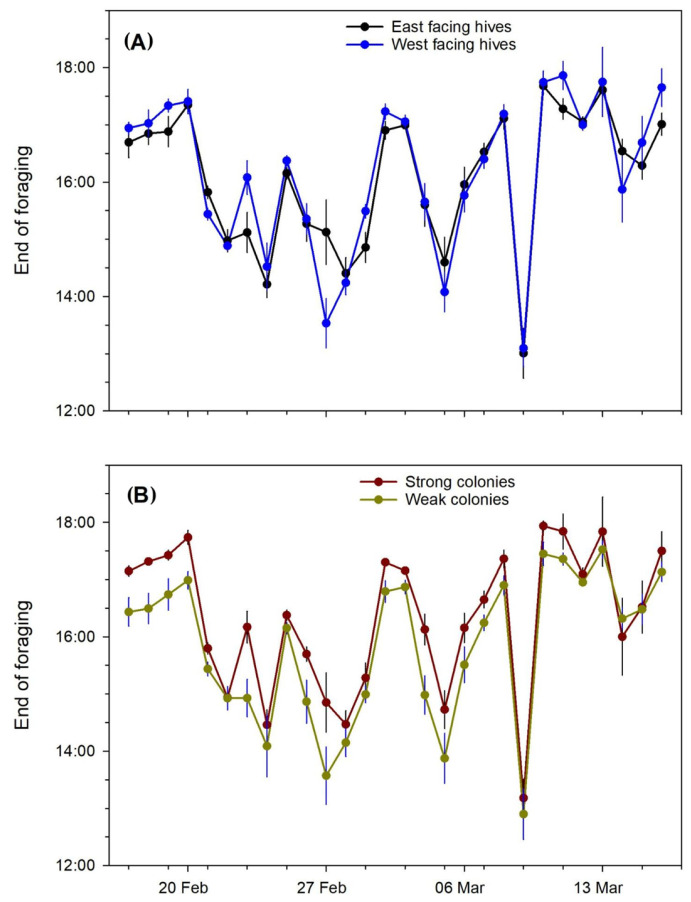
The average flight end time of the colonies during almond pollination for (**A**) east- vs. west-facing colonies and (**B**) strong vs. weak colonies. Error bars represent the standard error of the mean.

**Figure 6 insects-15-00112-f006:**
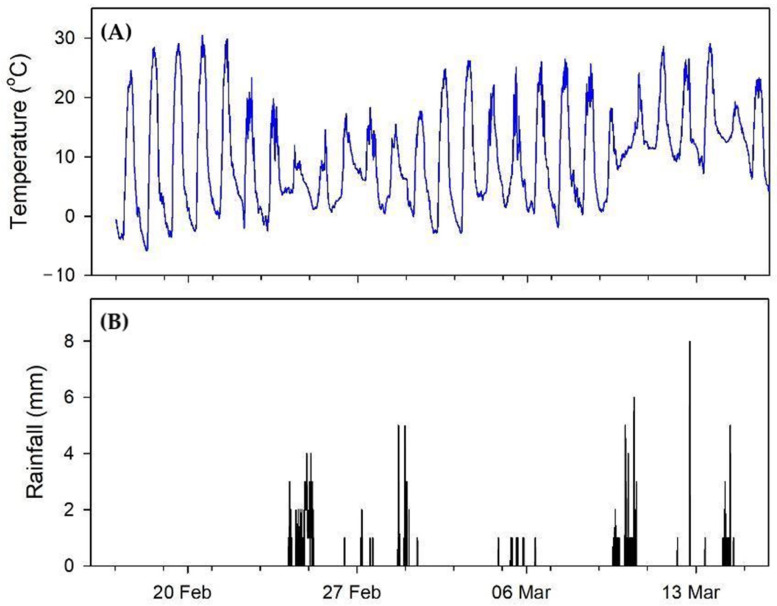
The weather conditions during the course of the study: (**A**) ambient temperature and (**B**) rainfall.

**Figure 7 insects-15-00112-f007:**
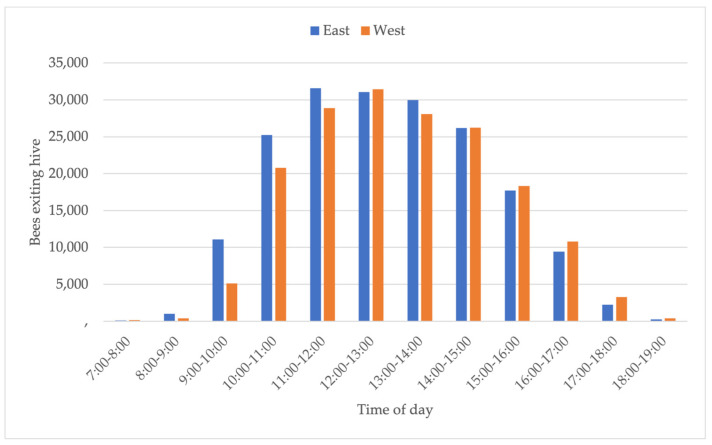
The average number of bee flights per hourly time interval in east- and west-facing groups.

**Figure 8 insects-15-00112-f008:**
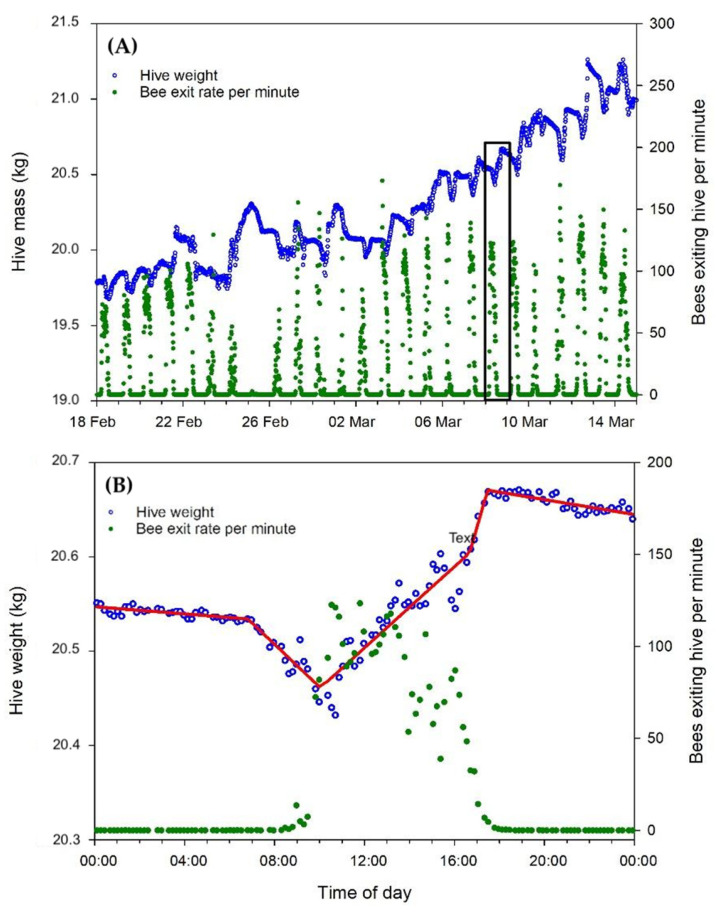
The hive mass and bee count data. (**A**) The sample data for a single colony throughout the experiment; (**B**) the 24 h dataset indicated in the black box outline in (**A**). The red line indicates the piecewise regression fit to the weight data.

**Table 1 insects-15-00112-t001:** The MANOVA results for the following factors: Direction (east or west), Strength (strong or weak), Day (the day of the year from 17 February to 15 March 2023) and all two-way interactions for bee colony foraging “start” times (the counter detects at least 10 bees per minute for the first time that day) and “stop” times (counter no longer detects at least 10 bees per minute for the rest of the day).

Response Variable	Fixed Effect	Num DF	Den DF	F Value	Pr > F
Start time	Direction	1	75.57	16.60	0.0001
	Strength	1	76.3	17.07	<0.0001
	Day	26	455	22.41	<0.0001
	Direction * strength	1	78.5	7.12	0.0093
	Direction * day	26	454.9	1.97	0.0034
	Strength * day	26	455.1	3.29	<0.0001
Stop time					
	Direction	1	83.53	0.02	0.8915
	Strength	1	84.56	25.94	<0.0001
	Day	26	453.5	53.31	<0.0001
	Direction * strength	1	85.93	1.59	0.2109
	Direction * day	26	453.3	2.20	0.0007

The asterisk (*) is used to indicate all main effects and interactions among the variables that it joins.

**Table 2 insects-15-00112-t002:** The MANOVA results for the following factors: Direction (east or west), Strength (strong or weak), Day (the day of the year from 17 February to 15 March 2023) and all two-way interactions for piecewise regression results. The “dawn break point”, “dusk break point” and “forager population”. See the text for the details of the response variables.

Response Variable	Fixed Effects	Num DF	Den DF	F Value	Pr > F
Dawn break point	Direction	1	161.9	1.61	0.2057
	Strength	1	159.8	0.00	0.9919
	Day	26	434.3	17.11	<0.0001
	Direction * strength	1	143.6	0.09	0.7590
	Direction * day	26	434.1	1.12	0.3139
	Strength * day	26	434.4	0.85	0.6796
Dusk break point	Direction	1	190.9	1.91	0.1686
	Strength	1	188.2	1.17	0.2814
	Day	26	435.8	6.61	<0.0001
	Direction * strength	1	165.2	2.30	0.1313
	Direction * day	26	435.6	0.93	0.5661
	Strength * day	26	436	0.78	0.7681
Forager population	Direction	1	117.3	14.85	0.0002
	Strength	1	116.2	7.21	0.0083
	Day	26	430.4	10.06	<0.0001
	Direction * strength	1	110	0.41	0.5235
	Direction * day	26	430.4	0.75	0.8046
	Strength * day	26	430.4	1.64	0.0256

The asterisk (*) is used to indicate all main effects and interactions among the variables that it joins.

## Data Availability

The datasets generated during the current study are available from the corresponding author upon reasonable request.

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
