# Peer review of "Hive Orientation and Colony Strength Affect Honey Bee Colony Activity during Almond Pollination"

_insects, 2024, doi:10.3390/insects15020112_

Round 1
Reviewer 1 Report
Comments and Suggestions for Authors
we recommend to the author change for another way to analyze statistically the data collected, and all parameters group of honeybees (3); cultivar (3); east or west facing (2), and year (minimum, two years of data collection) must be analyzed together.
to improve and obtain more reliable results, we recommend, at least two years collecting data;
Author Response
Authors: First of all, the Reviewer is quite right that a second field trial, while not strictly necessary, would have been ideal. However, apart from the expense in financial and human resources, we felt this dataset justified publication for two reasons: 1) We collected and compared data from two independent sensor types – gate counters and hive scales – which is rarely done in such field studies; and 2) These results largely confirmed data from a previous study, so at least in some respects this study represents a replication of that earlier study. We now state this in the text. Regarding the statistical design, first of all the almond varieties were all present in the same orchard, and so do not represent treatment groups. We are sorry this was not clear and have clarified that in the text. Secondly, as stated the honey bee queenlines were added to increase genetic diversity in the experimental groups. Queenline is often correlated with size (please see Danka, R.G.; Sylvester, H.A; Boykin, D. Environmental influences on flight activity of USDA-ARS Russian and Italian stocks of honey bees (Hymenoptera: Apidae) during almond pollination J. Econ. Entomol., 2006, 99(5), 1565-70 https://doi.org/ 10.1603/0022-0493-99.5.1565) among other factors and thus would have confounded the analysis, so we focused on a hive quality that is, we felt, easier to keep track of.
Reviewer 2 Report
Comments and Suggestions for Authors
insects-2796359 Reviewer comments
Manuscript insects-2796359: Hive orientation and colony strength affect honey bee colony activity during almond pollination
The manuscript is very interesting. The authors examined whether hive entrance orientation and hive strength influence the pollinating activity of honey bees in almond orchards. The authors have found that hives facing east have a marked advantage in starting their activity in the morning compared to hives facing west. Similarly, strong colonies show a significantly earlier start to activity and a longer work time of more than one hour than weak colonies. The practical implication of these findings lies in configuring the placement of colonies of varying strengths in a way that favors their foraging activity.
The uniqueness of the text is 90% by antiplagiarism.net
The experimental and statistical methods are correct.
The English is good. I did not find any misspellings.
There are some mistakes and comments.
1. Line 131—the authors wrote—is one of the Bee counter’s 24 tunnels. On Figure 1, the bee counter has 26 tunnels. Please check the correct number of tunnels.
2. Line 99: In this study data from entrance activity sensors (bee counters) - please add additional information and comparison with other bee-counting systems and add a citation (Son et al., 2019). Add to the references the following: Son, J.D.; Lim, S.; Kim, D.I.; Han, G.; Ilyasov, R.A.; Yunusbaev, U.B.; Kwon, H.W. Automatic bee-counting system with dual infrared sensor based on ICT. Journal of Apiculture (Korea) 2019, 34, 47–55, doi:10.17519/apiculture.2019.04.34.1.47.
3. Lines 114–124: The authors wrote about three study groups (Russian, Poll-line and Stock). I did not find any results about the comparison of these honeybee groups. Why did you use these honey bee groups? The authors should explain in the text.
4. What was the colony strength in each colony of groups (Russian, Poll-line and Stock) and in east-facing and west-facing directions in the beginning? While experimenting on Figure 3, colony strength increased more in the east-facing group. Why on Figure 4 did you compare strong and weak colonies? What colonies are included in the strong and weak groups? Are they grouped independently of their east-west orientation? Please explain in the text why you did this.
5. On Figure 8, what is the difference between hive weight and hive mass in kg? If there is no difference, please use similar. If they differ, please explain in the text.
6. Lines 346-348: The authors wrote: Furthermore, we have found that weak colonies, regardless of their orientation, started foraging 45 min later and cease foraging 38.3 min earlier compared to strong colonies, adding to an average of over an hour less foraging time per day. - The authors' explanation for this—the larger the bee population, the higher the probability for the individual bees to behave differently from the average bee, whether it is leaving or entering the hive—is not convincing to me. Please try to find an additional explanation for this.
7. The discussion part is a little weak; please add more discussion in comparison with other studies.
Please improve the manuscript according to the above comments.
Comments on the Quality of English Language
The English is well, minor editing of English language required
Author Response
Reviewer: The manuscript is very interesting. The authors examined whether hive entrance orientation and hive strength influence the pollinating activity of honey bees in almond orchards. The authors have found that hives facing east have a marked advantage in starting their activity in the morning compared to hives facing west. Similarly, strong colonies show a significantly earlier start to activity and a longer work time of more than one hour than weak colonies. The practical implication of these findings lies in configuring the placement of colonies of varying strengths in a way that favors their foraging activity.
Authors: The Reviewer has presented a good summary of the work.
Reviewer: The uniqueness of the text is 90% by antiplagiarism.net
The experimental and statistical methods are correct.
The English is good. I did not find any misspellings.
There are some mistakes and comments.
- Line 131—the authors wrote—is one of the Bee counter’s 24 tunnels. On Figure 1, the bee counter has 26 tunnels. Please check the correct number of tunnels.
Authors: Very sorry about the confusion. The Reviewer is correct – there were 26 tunnels.
- Line 99: In this study data from entrance activity sensors (bee counters) - please add additional information and comparison with other bee-counting systems and add a citation (Son et al., 2019). Add to the references the following: Son, J.D.; Lim, S.; Kim, D.I.; Han, G.; Ilyasov, R.A.; Yunusbaev, U.B.; Kwon, H.W. Automatic bee-counting system with dual infrared sensor based on ICT. Journal of Apiculture (Korea) 2019, 34, 47–55, doi:10.17519/apiculture.2019.04.34.1.47.
Authors: We had not seen that paper and have now refer to it as an example, along with another paper.
- Lines 114–124: The authors wrote about three study groups (Russian, Poll-line and Stock). I did not find any results about the comparison of these honeybee groups. Why did you use these honey bee groups? The authors should explain in the text.
Authors: This is a very good point, and one that was brought up by Reviewer 1. We used the expression “study groups” in our original submission, which led to this confusion, and we removed that expression. Briefly, queenline is often correlated with size (we now cite a reference) and to avoid confounding the analysis we focused on a hive quality that is easier to keep track of.
- What was the colony strength in each colony of groups (Russian, Poll-line and Stock) and in east-facing and west-facing directions in the beginning? While experimenting on Figure 3, colony strength increased more in the east-facing group. Why on Figure 4 did you compare strong and weak colonies? What colonies are included in the strong and weak groups? Are they grouped independently of their east-west orientation? Please explain in the text why you did this.
Authors: The reviewer is quite right that the east facing colonies showed marginally higher increase in strength over the course of the study, however this increase was not statistically significant and therefore was not considered further. Regarding the figures, we focused on the main effects (hive orientation and colony strength) rather than the interaction term (orientation x strength), partly because we think those of primary interest and partly to render the graphs easier to interpret. The main reason for the comparison of strong and weak colonies (in Figure 4) is due to the economic consideration for almond pollination price per hive. Two aspects of colony strength form a foundation of almond pollination contracts: minimum colony strength (8 FOB as examined in the field by rapid inspection) and a bonus for strong colonies (usually those in excess of something like 12 FOB). We now explain this in the Introduction. As stated in the text, we grouped the bees into weak and strong based on the Liebefeld method, and the criterion for Strong and Weak was in some respects simply a function of the sizes of the colonies as we received them. In the field colonies were grouped independently of either strength or orientation and fortunately for the experimental design both factors were about evenly distributed.
- On Figure 8, what is the difference between hive weight and hive mass in kg? If there is no difference, please use similar. If they differ, please explain in the text.
Authors: Thank you. That was a mistake in the figure, which we have now corrected.
- Lines 346-348: The authors wrote: Furthermore, we have found that weak colonies, regardless of their orientation, started foraging 45 min later and cease foraging 38.3 min earlier compared to strong colonies, adding to an average of over an hour less foraging time per day. - The authors' explanation for this—the larger the bee population, the higher the probability for the individual bees to behave differently from the average bee, whether it is leaving or entering the hive—is not convincing to me. Please try to find an additional explanation for this.
Authors: We regret that the Reviewer does not find that explanation convincing. Indeed, we do not find it particularly satisfying either. However, that is the simplest explanation and so the most likely and the one we would need to eliminate before considering more complicated explanations. The reason is that the criterion for the commencement of foraging activity, 10 bees per minute exiting the hive, is not a function of colony size. Let us consider two colonies, in all respects identical except that one is twice the size of the other. We assume the same proportion of each colony departs for foraging, that the period of time over which foragers exit is the same, and the mean time of forager exits is the same for both colonies. In that case the larger colony will very likely hit the criterion of 10 bees per minute before the smaller colony, simply because there are more bees to exit the hive. Of course, we made assumptions but they are reasonable assumptions. We tried to clarify this further in the text.
- The discussion part is a little weak; please add more discussion in comparison with other studies.
Authors: We have added material to the Discussion, and we added two references with studies involving bee counters, so we hope we have improved the paper on that count. There are, as we pointed out, rather few studies comparing bee counter data with hive scale data.
Reviewer 3 Report
Comments and Suggestions for Authors
General comments
Research on pollination services is of high interest to stakeholders in the agricultural community. The fragility of the Apis-almond system due its reliance on one species of pollinator and the pressure of many stressors (pathogens, pesticides, fungicides, climate change, invasives, reduction in forage resources for Apis off season) contribute to Apis colony declines. Hence research to improve foraging and colony strength is needed and of interest.
Regarding the statistical methods used: What statistical test was used to compare the FOBs in the two groups? Using Excel would not be considered robust enough for publication. JMP or SAS or some equivalent would be better for publication.
This paper builds on the work of an earlier paper Meikle (2022). In this earlier paper by Meikle (2022) which investigates south, north, east and west entrance orientations the sample size of 3 hives for each orientation is quite low and below an experimental sample size of 10 which is usually a minimum for doing statistical analysis to determine if there is a significant difference between treatments.
Comment of nest orientation: In the literature on native bee nest site selection has documented that solitary bees significantly choose a south-facing slope in western landscapes.
Regarding foraging measurements: It seems that there are several ways this can be looked at. Is foraging earlier better (getting to the resource first), foraging for more hours in the day, foraging with more workers at any given time period better? What is the best measurement? Should each of these be tested against each other?
Comments on the Quality of English Language
Comments on English editing
Simple Summary
Line 15: delete “and”
Abstract
Line 27: delete “also”
Discussion
Lines 373 and 375: these two phrases contradict each other? Are both opinions cited in the same paper?
Author Response
General comments
- Research on pollination services is of high interest to stakeholders in the agricultural community. The fragility of the Apis-almond system due its reliance on one species of pollinator and the pressure of many stressors (pathogens, pesticides, fungicides, climate change, invasives, reduction in forage resources for Apis off season) contribute to Apis colony declines. Hence research to improve foraging and colony strength is needed and of interest.
Regarding the statistical methods used: What statistical test was used to compare the FOBs in the two groups? Using Excel would not be considered robust enough for publication. JMP or SAS or some equivalent would be better for publication.
Authors: For basic statistics such as t tests and simple correlations we feel even a pencil and paper (as one co-author is old enough to remember using) is sufficient. We removed the reference to Excel.
- This paper builds on the work of an earlier paper Meikle (2022). In this earlier paper by Meikle (2022) which investigates south, north, east and west entrance orientations the sample size of 3 hives for each orientation is quite low and below an experimental sample size of 10 which is usually a minimum for doing statistical analysis to determine if there is a significant difference between treatments.
Authors: The Reviewer is correct in that sample sizes in Meikle et al. (2022) were low (although that study considered all 4 cardinal directions) and the sample sizes per treatment group presented here were higher and therefore more convincing with respect to East-West orientation.
- Comment of nest orientation: In the literature on native bee nest site selection has documented that solitary bees significantly choose a south-facing slope in western landscapes.
Authors: We did not review the literature concerning the nesting habits of solitary bees as their life cycles are not comparable to those of social honey bees, firstly as the vast majority of solitary bees nest in the ground and secondly, solitary bees of temperate regions have seasonal cycles and do not maintain active nests throughout the year as is the case with honey bees. We realise that we did not make a clear distinction in the text when citing literature on feral honey bee nests, as opposed to any other wild bee species, and we have now addressed that.
- Regarding foraging measurements: It seems that there are several ways this can be looked at. Is foraging earlier better (getting to the resource first), foraging for more hours in the day, foraging with more workers at any given time period better? What is the best measurement? Should each of these be tested against each other?
Authors: These are very good points and we try now to address them in the Discussion. We now state “Also, we examined foraging parameters chosen at least in part for their ease in computation and comparison across sensor types. Other parameters, such as the length and distribution of the total foraging effort, may also be of interest in future work.”
- Line 15: delete “and”
Authors: Done
- Abstract. Line 27: delete “also”
Authors: Done
- Discussion. Lines 373 and 375: these two phrases contradict each other? Are both opinions cited in the same paper?
Authors: While those phrases seem contradictory, they are not necessarily so. The key word is “proportionately”. Large colonies may have more foragers overall, and may send more foragers than smaller colonies to abundant resources, but not necessarily “proportionately” more. More foragers go simply because there are more foragers in absolute numbers. We tried to clarify that.
Round 2
Reviewer 1 Report
Comments and Suggestions for Authors
The changes made by the authors as well as the inclusion of some references improved the manuscript. However, we maintain our opinion that the statistical analysis should be different, as we have the variables year, cultivar (3), west or east facing (2), group of bees (3) that were not considered in analysis. Furthermore, the other main journal is demanding at least two years of collecting data.
Author Response
Reviewer 1: The changes made by the authors as well as the inclusion of some references improved the manuscript. However, we maintain our opinion that the statistical analysis should be different, as we have the variables year, cultivar (3), west or east facing (2), group of bees (3) that were not considered in analysis. Furthermore, the other main journal is demanding at least two years of collecting data.
Authors: Thank you for recognising the improvements to the manuscript. With regard to statistical analysis we can try to explain with more clarity our rationale:
- As we stated in our first response, almond cultivar (3) does not represent a treatment as all single almond orchards are composed of 2-4 cultivars for purposes of cross pollination. For cultivars to be considered as a variable it would be necessary to run the study in distinct orchards with different cultivar combinations, which was not done here and in any case not how commercial almonds are typically grown.
- Groups of bees (queen lines here) are indeed of significant interest and we plan to perform future studies with respect to this variable. However, in this study the basis for adding queen lines was for the benefit of genetic diversity and due to low sample size (8 each) it did not offer a robust data set for this variable.
- While we continue to agree that the two year data collection would be better we are not in a position to provide this. However, we believe that evaluating two different and independent sensors for each hive is sufficiently novel to help compensate for the lack of a second season.
Reviewer 3 Report
Comments and Suggestions for Authors
Hive orientation and colony strength affect honey bee colony activity during almond pollination
Sandra Kordić Evans * , Huw Evans , William G. Meikle , George Clouston
Second Comment round, Jan 9, 2023
1.Authors: For basic statistics such as t tests and simple correlations we feel even a pencil and paper (as one co-author is old enough to remember using) is sufficient. We removed the reference to Excel.
The authors response to initial comments to use a scientifically acceptable statistically program rather than Excel was dismissive. They simply removed explaining which statistical program used for analysis of "data were analysed using correlation and two-tailed t-test functions" (line 177-178).
This is not acceptable. Excel is a spreadsheet program and is not generally acceptable to journals for performing statistical analysis. As I suggested there are several easily available packages that can do these simple tests and the software you use needs to be included in the methods section not just omitted. Try SPSS, R, SAS, JMP. These programs are available for $20 through a university or free like R. There are references in Google Scholar on why Excel is not used for statistical analysis.
Response 2
Authors: The Reviewer is correct in that sample sizes in Meikle et al. (2022) were low (although that study considered all 4 cardinal directions) and the sample sizes per treatment group presented here were higher and therefore more convincing with respect to East-West orientation.
The point I was making is that you eliminated testing two of the “cardinal directions” south and north based on a study with very low sample sizes. All four directions should have been included in this study with increased sample sizes to see if the results were the same with higher samples sizes of at least 10. A higher sample size of more than 10 is even better. The weakness of the current study is that you are relying completely on an earlier study with too low a sample size for an experimental study. You had an opportunity to correct this by including north and south in this study but you did not. You also did not state that you were going to test north and south again in the future with a higher sample size. Thus you cannot suggest that east is superior without further testing with a sample size of three.
Response 3
Authors: We did not review the literature concerning the nesting habits of solitary bees as their life cycles are not comparable to those of social honey bees, firstly as the vast majority of solitary bees nest in the ground and secondly, solitary bees of temperate regions have seasonal cycles and do not maintain active nests throughout the year as is the case with honey bees. We realise that we did not make a clear distinction in the text when citing literature on feral honey bee nests, as opposed to any other wild bee species, and we have now addressed that.
I was not suggesting that you include a review of the solitary bee nesting habits. Again, you missed the point. The point was that I calling your attention to the biological behavior of other bee species which prefer to orient their nest entrances facing south. This was just a gift of knowledge that I was passing on from the bee research sages (Thorp, Rozen, Cane). Yes, solitary bees are different from social bees but they too have to acquire pollen and nectar resources just like honey bees but in a condensed period of time (weeks, months) and without workers.
- Regarding foraging measurements: It seems that there are several ways this can be looked at. Is foraging earlier better (getting to the resource first), foraging for more hours in the day, foraging with more workers at any given time period better? What is the best measurement? Should each of these be tested against each other?
Authors: These are very good points and we try now to address them in the Discussion. We now state “Also, we examined foraging parameters chosen at least in part for their ease in computation and comparison across sensor types. Other parameters, such as the length and distribution of the total foraging effort, may also be of interest in future work.”
- Discussion. Lines 373 and 375: these two phrases contradict each other? Are both opinions cited in the same paper?
Authors: While those phrases seem contradictory, they are not necessarily so. The key word is “proportionately”. Large colonies may have more foragers overall, and may send more foragers than smaller colonies to abundant resources, but not necessarily “proportionately” more. More foragers go simply because there are more foragers in absolute numbers. We tried to clarify that.
Line 399-400: “and because these results largely confirmed those from a previous study, so in some respects this study represents a replication of that earlier study.”
Comment: This statement is not accurate because this study only tested two directions not all four so it did not “replicate the earlier study”, replicate means make an exact copy. I suggest this be edited.
Because you did not re-test north and south with adequate sample sizes and do not discuss this deficiency in the discussion, I am not sure what you can say about this study.
Author Response
Reviewer 3: The authors response to initial comments to use a scientifically acceptable statistically program rather than Excel was dismissive. They simply removed explaining which statistical program used for analysis of "data were analysed using correlation and two-tailed t-test functions" (line 177-178).
This is not acceptable. Excel is a spreadsheet program and is not generally acceptable to journals for performing statistical analysis. As I suggested there are several easily available packages that can do these simple tests and the software you use needs to be included in the methods section not just omitted. Try SPSS, R, SAS, JMP. These programs are available for $20 through a university or free like R. There are references in Google Scholar on why Excel is not used for statistical analysis.
Authors: Thank you for the explanation and guidance, we have now analyze the data with a repeated measures MANOVA using SAS, as we do with the other analyses, and this is now reflected in the Methods and Materials. We also removed Table 1, since it did not show significant differences.
Reviewer 3: The point I was making is that you eliminated testing two of the “cardinal directions” south and north based on a study with very low sample sizes. All four directions should have been included in this study with increased sample sizes to see if the results were the same with higher samples sizes of at least 10. A higher sample size of more than 10 is even better. The weakness of the current study is that you are relying completely on an earlier study with too low a sample size for an experimental study. You had an opportunity to correct this by including north and south in this study but you did not. You also did not state that you were going to test north and south again in the future with a higher sample size. Thus you cannot suggest that east is superior without further testing with a sample size of three.
Authors: Thank you for clarifying your concern, we agree that assessing all four cardinal point with at least 10 samples in each group would have given a more robust data set. As in all field studies there were limitations, and the design for this study was determined carefully based on the resources we had, specifically the number of hives and accompanying equipment (24). Had we placed hives facing in all four cardinal directions, we would have come up once again short in terms of sample size, by the Reviewer’s own standard. Given that we would use only two treatment groups, the only question remaining was whether it should be east/west or north/south. The answer to this was partly dictated by the geography of the almond orchard we had access to - bee hives in almond pollination are usually placed on pallets and oriented in such a way not to point across access roads - and partly dictated by previous work (Meikle et al. 2023 https://doi.org/10.1080/00218839.2023.2165769), which indicated that an east-facing aspect appeared to advantageous. While this study included only two cardinal directions, these results can be compared to East/West pairwise comparisons in the Meikle et al. (2023) paper.
We have now modified the manuscript (line 305) to explain east orientation being a preferable only when compared to west orientation and to suggest further study on other cardinal points. Indeed to claim an optimum orientation a study should consider all reasonable configurations including south-east, south-west, north-east and north-west (line 396).
Reviewer 3: I was not suggesting that you include a review of the solitary bee nesting habits. Again, you missed the point. The point was that I calling your attention to the biological behavior of other bee species which prefer to orient their nest entrances facing south. This was just a gift of knowledge that I was passing on from the bee research sages (Thorp, Rozen, Cane). Yes, solitary bees are different from social bees but they too have to acquire pollen and nectar resources just like honey bees but in a condensed period of time (weeks, months) and without workers.
Authors: Apologies for the misunderstanding and thank you for your patience and the valuable references.
Reviewer 3: Line 399-400: “and because these results largely confirmed those from a previous study, so in some respects this study represents a replication of that earlier study.” Comment: This statement is not accurate because this study only tested two directions not all four so it did not “replicate the earlier study”, replicate means make an exact copy. I suggest this be edited.
Authors: Thank you, we have now edited this inaccuracy.
Reviewer 3: Because you did not re-test north and south with adequate sample sizes and do not discuss this deficiency in the discussion, I am not sure what you can say about this study.
Authors: Having been through the review process we now understand better why the choice of two cardinal points is seen as problematic and have now addressed this in the discussion. We are not claiming that east orientation is optimum, rather, that it affects bee foraging activity favourably when compared to west orientation (line 305 correction above), and possibly more interestingly, how hive strength compounds this effect.